# Extracellular Vesicle Membrane Protein Profiling and Targeted Mass Spectrometry Unveil CD59 and Tetraspanin 9 as Novel Plasma Biomarkers for Detection of Colorectal Cancer

**DOI:** 10.3390/cancers15010177

**Published:** 2022-12-28

**Authors:** Srinivas Dash, Chia-Chun Wu, Chih-Ching Wu, Sum-Fu Chiang, Yu-Ting Lu, Chien-Yuh Yeh, Jeng-Fu You, Lichieh Julie Chu, Ta-Sen Yeh, Jau-Song Yu

**Affiliations:** 1Graduate Institute of Biomedical Sciences, College of Medicine, Chang Gung University, Taoyuan 33302, Taiwan; 2Molecular Medicine Research Center, Chang Gung University, Taoyuan 33302, Taiwan; 3Department of Medical Biotechnology and Laboratory Science, College of Medicine, Chang Gung University, Taoyuan 33302, Taiwan; 4Department of Otolaryngology-Head and Neck Surgery, Chang Gung Memorial Hospital, New Taipei City 33305, Taiwan; 5Division of Colon and Rectal Surgery, Chang Gung Memorial Hospital, New Taipei City 33305, Taiwan; 6School of Traditional Chinese Medicine, Chang Gung University, Taoyuan 33302, Taiwan; 7Liver Research Center, Chang Gung Memorial Hospital, New Taipei City 33305, Taiwan; 8Department of Surgery, Chang Gung Memorial Hospital, Linkou & Chang Gung University, New Taipei City 33305, Taiwan; 9College of Medicine, Chang Gung University, Taoyuan 33302, Taiwan; 10Research Center for Food and Cosmetic Safety, College of Human Ecology, Chang Gung University of Science and Technology, Taoyuan 33302, Taiwan

**Keywords:** colorectal cancer, extracellular vesicles, membrane protein, AIMS, iTRAQ, targeted mass spectrometry, CD59, TSPAN9

## Abstract

**Simple Summary:**

It is well recognized that EVs carry many membrane proteins on their surface that can transduce signals to recipient cells and serve as biomarkers/target proteins for the diagnosis /treatment of diseases. Although numerous studies have reported profiling of EV proteins from tumor cells, few efforts have been made to systemically explore tumor cell-derived EV membrane proteins and verify that these membrane proteins are useful plasma biomarkers. The present study explored the EV membrane protein profile in common among CRC cell lines and assessed alterations in the plasma EV proteome of CRC patients compared with healthy subjects. It further identified the plasma EV membrane proteins, CD59 and TSPAN9, as a novel biomarker panel that effectively detected CRC through targeted MS-based quantification of roughly a dozen selected candidate membrane proteins in plasma EV samples from 73 CRC patients and 80 healthy subjects.

**Abstract:**

Extracellular vesicles (EVs) are valuable sources for the discovery of useful cancer biomarkers. This study explores the potential usefulness of tumor cell-derived EV membrane proteins as plasma biomarkers for early detection of colorectal cancer (CRC). EVs were isolated from the culture supernatants of four CRC cell lines by ultracentrifugation, and their protein profiles were analyzed by LC-MS/MS. Bioinformatics analysis of identified proteins revealed 518 EV membrane proteins in common among at least three CRC cell lines. We next used accurate inclusion mass screening (AIMS) in parallel with iTRAQ-based quantitative proteomic analysis to highlight candidate proteins and validated their presence in pooled plasma-generated EVs from 30 healthy controls and 30 CRC patients. From these, we chose 14 potential EV-derived targets for further quantification by targeted MS assay in a separate individual cohort comprising of 73 CRC and 80 healthy subjects. Quantitative analyses revealed significant increases in ADAM10, CD59 and TSPAN9 levels (2.19- to 5.26-fold, *p* < 0.0001) in plasma EVs from CRC patients, with AUC values of 0.83, 0.95 and 0.87, respectively. Higher EV CD59 levels were significantly correlated with distant metastasis (*p* = 0.0475), and higher EV TSPAN9 levels were significantly associated with lymph node metastasis (*p* = 0.0011), distant metastasis at diagnosis (*p* = 0.0104) and higher TNM stage (*p* = 0.0065). A two-marker panel consisting of CD59 and TSPAN9 outperformed the conventional marker CEA in discriminating CRC and stage I/II CRC patients from healthy controls, with AUC values of 0.98 and 0.99, respectively. Our results identify EV membrane proteins in common among CRC cell lines and altered plasma EV protein profiles in CRC patients and suggest plasma EV CD59 and TSPAN9 as a novel biomarker panel for detecting early-stage CRC.

## 1. Introduction

Colorectal cancer (CRC) is the third-most common cancer overall and the second-most deadly, with an incidence rate of two million new CRC cases and one million deaths per year, accounting for approximately 1 in every 10 case of cancer and death. Developed countries have a 4-fold greater incidence rate than developing countries; however, because fatality rates are higher in developing countries, mortality rates are less variable [1,2]. According to the US Centers for Disease Control and Prevention (CDC), those diagnosed with localized colorectal cancer have a 5-year survival probability of 91%; this compares with 70% for regional-stage cancer and 11% for distant-stage disease. Data from the CDC show that CRC incidence and mortality have decreased in recent years as a result of screening initiatives [3]. Although colonoscopies and biopsies have greatly improved the diagnosis of CRC, they are extremely invasive procedures [4,5,6,7]. The discovery of molecular signals in bodily fluids could lead to the development of ’liquid biopsies,’ which would solve many of the problems associated with traditional tissue sampling (e.g., invasiveness, tumor heterogeneity). Liquid biopsies are non-invasive and have come to represent the pinnacle of good screening materials for stage-specific diagnosis and prognosis and assessment of response to medication. One of the most appealing aspects of human fluids for this purpose is that they enable early disease detection. A variety of physiological fluid samples have been used for both discovery and validation of putative biomarkers, including serum, plasma, urine, cerebrospinal fluid, saliva, ascites, and amniotic fluid [8,9,10,11]. 

Because of the genetic variability of CRC, many researchers agree that a panel of biomarkers will be required to achieve sufficient sensitivity for clinical application as a screening biomarker [12,13,14,15]. Immunoassays are used to quantify possible biomarkers of interest in control and disease samples, with enzyme-linked immunosorbent assay (ELISA) being the gold standard. Current methods are constrained by signal detection limits of clinical indicators, which together with the paucity of well-characterized, highly specific monoclonal antibodies has limited existing techniques [16,17]. Research on extracellular vesicles (EVs) has renewed interest in biomarkers by virtue of the ability of EVs to potentially overcome these highlighted limitations in biomarker identification during the discovery phase through proteomic techniques. Among recent discoveries in the CRC field, the potential of EVs as a source of biomarkers has been a source of enthusiasm for academics around the world. EVs formed from body fluids could be a particularly appealing source of biomarkers in this regard, as these vesicles are expected to reflect the molecular composition of the cells that secrete them [18,19,20].

EVs are small, membrane-bound vesicles ranging in size from nanometers to micrometers in diameter that are secreted by multiple cell types [21,22,23,24]. EVs encapsulate many different proteins, nucleic acids, and other macromolecules, depending on the parental cells and the body’s pathophysiological status [25]. Among the crucial functions of EVs are intercellular trafficking and communications [26,27]. They have also been linked to innate and adaptive immune responses, and have been shown to aid angiogenesis and metastasis [28,29]. The ultimate goal of EV-based biomarker analysis is to develop biomarkers for early diagnosis and prognosis of pathologies. Recent research has found that the quantities of EVs and/or the composition of EV membrane proteins in body fluids from healthy people and cancer patients might differ significantly [30,31,32,33], and that some tumor EV membrane proteins can predict organotropic metastasis [34,35,36]. Because of cargo sorting into EVs, the concentration of biomarkers near the EV source aids in the detection of relatively modestly expressed biomarkers that would otherwise go undetected. 

Membrane proteins, encompassing all cell surface, peripheral and integral proteins, are of special interest among potential EV biomarkers as they are exposed on the vesicle membrane and hence are more easily detected by appropriate affinity reagents (e.g., antibodies, aptamers) [37,38,39]. A variety of EV membrane proteins have been tested as serum/plasma biomarkers and shown promise in detecting different types of cancer. These include CD147/basigin combined with CD9, and epithelial cell adhesion molecule (EpCAM) combined with CD63 for CRC [40,41]; CA-125, EpCAM and CD24 combined with CD9 for ovarian cancer [42]; EpCAM, prostate-specific membrane antigen (PSMA) and ephrin A2 for prostate cancer [43,44]; carcinoembryonic antigen (CEA), GPC-3 and PD-L1 for hepatocellular carcinoma [45]; epidermal growth factor receptor (EGFR), EpCAM, glypican 1 (GPC1) and EPH receptor A2 (EphA2) for pancreatic cancer [46]; and CA153, CA125, CEA, human epidermal growth factor receptor 2 (HER2), EGFR, PSMA and EpCAM for metastatic breast cancer [47]. These encouraging data indicate the high potential of EV membrane proteins as targets for the development of novel liquid biopsy assays for cancer detection. However, very few studies have directly investigated the EV membrane protein profile of specific cancer types, prioritized candidate proteins, and verified their potential clinical utility.

Mass spectrometry (MS) has been shown to be a useful tool for detecting EV-derived biomarkers [48,49,50]. Before committing resources to configuring quantitative assays, it is critical to choose from among the enormous number of early candidates and identify those that are most likely to be useful as a blood-based marker. Because many candidates come from discovery trials performed on tissue (or proximal fluids), and the number of such candidates is potentially unwieldy, the approach utilized should be capable of determining the presence of each candidate in plasma. Accurate Inclusion Mass Screening (AIMS), a type of focused mass spectrometry, is extremely useful in this context and provides a framework for unbiased discovery and subsequent targeted MS-based quantification [51]. AIMS analysis takes advantage of the high resolution and mass accuracy of the Orbitrap mass spectrometer to target individual ions (peptides) of interest. iTRAQ (isobaric tags for relative and absolute quantitation), an untargeted quantitative proteomics technique [52], paired with liquid chromatography-tandem mass spectrometry (LC-MS/MS) analysis has become a potent quantitative proteomic methodology because of its high-throughput setup, high sensitivity, and enhanced accuracy, offering significant benefits compared with previous proteomics approaches [53,54,55]. 

In this study, we used 2D-LC-MS/MS to perform an in-depth analysis of the protein profile of EVs prepared from culture supernatants of four CRC cell lines, ultimately identifying 518 EV membrane proteins expressed in common among at least three CRC cell lines via bioinformatics analysis. The presence of candidate membrane proteins in plasma-derived EVs was validated by LC-MS/MS utilizing the AIMS mode and iTRAQ-based quantification in clinical samples pooled from 30 healthy controls (HCs) and 30 CRC patients. We then chose 56 prospective targets from AIMS and iTRAQ experiments and divided them into three categories (tier 1 to 3) according to their relative presence in the CRC group and the relative abundance between HC and CRC groups. A total of 13 targets from the tier 1 category and the EV marker, CD19, were further selected for a verification study on plasma EV samples from 80 HC subjects and 73 CRC patients. For this verification study, we performed high-throughput, focused measurements of the selected target proteins with high precision and sensitivity by implementing targeted MS assay runs in product ion scanning (PIS) MS/MS mode [56]. The results of this quantitative analysis suggest that a biomarker panel composed of plasma EV CD59 and TSPAN9 (tetraspanin 9) can efficiently distinguish CRC patients from HCs with an AUC (area under the receiver operating characteristic [ROC] curve) value of 0.98. Plasma EV CD59 and TSPAN9 individually also fared equally well, with AUC values of 0.95 and 0.87, respectively. In addition, these markers proved to be outstanding diagnostic markers for stage I and II CRC patients, with an AUC of 0.99 in tandem and individual AUC values of 0.96 and 0.86 for CD59 and TSPAN9, respectively.

## 2. Materials and Methods

### 2.1. Clinical Specimens 

Peripheral blood samples were collected from healthy donors and histologically confirmed surgery-naive CRC patients at the Department of Colon and Rectal Surgery (Chang Gung Memorial Hospital, Taoyuan, Taiwan) from 2016 to 2020. Inclusion criteria were (1) CRC patients with definite diagnosis; (2) CRC patients receiving standard treatments including surgery and/or chemotherapy as the protocol; (3) Over 20 years old; (4) Adults, as the control group, aged 20 years or older who were confirmed to have no CRC by colonoscopy. Exclusion criteria were (1) Those who were under 20 years old or legally disabled; (2) There was a history of other cancers. We did not exclude patients who received chemotherapy within 30 days before surgery, but in general, elective surgery would be avoided within 30 days after chemotherapy to prevent post-operative side effects. Moreover, all patients enrolled in this study did not die within 30 days after surgery. For preparation of plasma samples, blood was collected in EDTA tubes (10 mL per subject) and incubated at room temperature for up to 30 min, followed by centrifugation at 2000× *g* for 10 min at room temperature to obtain a cell pellet. The cell pellet was discarded to remove blood cells. Samples were aliquoted (1.0 mL) in sterile cryotubes and immediately frozen at −80 °C for storage until use. The discovery cohort consisted of 30 healthy individuals and 30 CRC patients. The plasma samples from all subjects were pooled into two groups for AIMS and iTRAQ-based LC-MS/MS quantitation. For the verification study via product ion scanning MS, plasma samples from 80 healthy subjects and 73 CRC patients were included and EVs were isolated individually from both controls and patients. All CRC patients had histologically verified adenocarcinoma. None of the CRC patients from stage 1 to stage 3 received chemotherapy before surgery. On the other hand, among 23 stage 4 CRC patients (7 patients in the discovery group, 16 patients in the verification group), 5 received chemotherapy before surgery and achieved the down-staging effect. Therefore, they received surgery for tumor resection, but blood sampling was still done before surgery. Patient characteristics were retrieved from clinical and pathology records, including gender, age, tumor location, histological grade, tumor stage, CEA level, preoperative laboratory data, operation date, operation method, tumor recurrence, follow-up date, and follow-up status. All patients were subjected to a follow-up strategy involving regular outpatient visits, CEA tests every 3 to 6 months, colonoscopy every 1 to 2 years, and imaging studies (chest X-ray and liver sonography or computed tomography) every year. The characteristics of the study subjects for both the discovery and verification phases are summarized in Appendix A. All individuals provided informed consent for blood donation according to the guidelines approved by the Institutional Review Board (IRB No. 201601848B0 and 201801888B0).

### 2.2. Cell Cultures

Human CRC cell lines (HT29, SW480, Colo205 and SW620) were obtained from the American Type Culture Collection. HT29 and Colo205 cells were cultured in RPMI-1640 medium (Invitrogen Corporation, Carlsbad, CA, USA) and SW480 and SW620 cells in Leibovitz’s L-15 medium (Gibco, Grand Island, NY, USA) supplemented with 10% heat-inactivated fetal bovine serum (Invitrogen), 100 unit’s/mL penicillin, and 100 mg/mL streptomycin. All cultures were incubated in humidified air at 5% CO_2_ and 37 °C.

### 2.3. EV Isolation from CRC Cell Lines

Confluent (70–80%) cells from all cell lines cultured in 15 cm dishes were washed twice with phosphate-buffered saline (PBS) and starved in serum-depleted medium (20 mL). After 24 h of incubation, the conditioned medium was centrifuged at 1000× *g* for 5 min, followed by another centrifugation step at 2000× *g* for 10 min, and cellular debris discarded. The supernatant was transferred to ultracentrifuge tubes for centrifugation at 10,000× *g* for 30 min. The medium was filtered using a 0.2 µm pore syringe filter (6786-1302, GE Healthcare, Chicago, IL, USA). A proportion of the filtered medium (1 mL) was stored for NTA and the remainder ultra-centrifuged at 100,000× *g* for 2 h at 4 °C using a SW32Ti rotor (Optima XE, Beckmann Coulter, Brea, CA, USA). EV pellets were washed with 10 mL PBS, followed by a second step of ultracentrifugation (SW41Ti rotor) at 100,000× *g* for 2 h at 4 °C. The supernatant was discarded, and the pellet resuspended in 200 µL PBS for functional assays. EVs used for protein extraction were resuspended in 250 µL of lysis buffer. The isolated EVs were stored at −80 °C for experimental use.

### 2.4. EV Isolation from Human Plasma Samples

Plasma samples (1 mL plasma per case) were diluted 10-fold with PBS, transferred to 50 mL tubes and centrifuged sequentially at 500× *g* for 5 min and 2000× *g* for 15 min. The supernatant was transferred to ultracentrifuge tubes and further centrifuged for 30 min at 10,000× *g*. The supernatant fraction obtained was collected and centrifuged for 2 h at 100,000× *g*. After discarding the supernatant from this step, the pellet was dissolved in PBS, passed through a 0.2 µm filter, collected in fresh tubes, and further centrifuged for 90 min at 100,000× *g*. The resulting pellet was resuspended in 200 µL PBS and stored at −80 °C. All centrifugation steps were performed at 4 °C using a SW41Ti rotor (Optima XE, Beckmann Coulter).

### 2.5. Transmission Electron Microscopy

Isolated EVs were resuspended in 4% paraformaldehyde (50–100 µL) and 10 µL aliquots were deposited on Formvar/carbon coated EM grids. The grid was covered and membrane adsorption conducted for 20 min in a dry environment. The grids (membrane side down) were transferred to drops of PBS (100 µL) with clean forceps for washing, followed by retransfer to a 50 µL drop of 1% glutaraldehyde for 5 min. Next, the grid was washed with distilled water 8 times for 2 min each, contrasted with a 50 µL drop of uranyl acetate solution for 5 min, and embedded with 50 µL methyl cellulose-UA for 10 min on ice. After removal of the grid using stainless steel loops, excess fluid was blotted and air-dried. The grid was examined under an electron microscope (JEM 1230, JEOL Ltd., Tokyo, Japan) at 80 kV.

### 2.6. Nanoparticles Tracking Analysis (NTA) 

Absolute size distribution of EVs was analyzed using Nano Sight LM10 with NTA3.2 software (Nano Sight Ltd., Minton Park, UK) for both data acquisition and analysis. Particles were automatically tracked and sized based on Brownian motion and the diffusion coefficient. Filtered PBS was used as control and blank samples. The NTA measurement conditions were as follows: temperature, 24.0 ± 0.5 °C; viscosity, 0.99 ± 0.01 cP; frames per second, 25; measurement time, 60 s. The detection threshold was similar in all samples. Three recordings were performed for each sample.

### 2.7. Flow Cytometry Analysis of EV-Bound Beads

EVs were attached to 4 µm aldehyde/sulfate latex beads (Invitrogen) by mixing 30 µg EVs in 10 µL beads for 15 min at room temperature with continuous rotation. The suspension was diluted to 1 mL with PBS and incubated overnight with continuous rotation at 4 °C. The reaction was terminated with 100 mM glycine and maintained for 30 min at room temperature. Next, the solution was centrifuged for 3 min at 4000 rpm in a microcentrifuge and the resulting supernatant discarded. Beads were washed twice with PBS/0.5% BSA with centrifugation at 4000 rpm for 3 min. EV-bound beads were incubated with biotin-labeled anti-CD9 (catalog number: 13-0098, eBioscience, San Diego, CA, USA) for 30 min with continuous rotation at 4 °C and rewashed twice with PBS/0.5% BSA with centrifugation at 4000 rpm for 3 min. Next, beads were incubated with streptavidin conjugated phycoerythrin (SA-PE) (BioLegend) for 30 min with continuous rotation at 4 °C. Beads were washed three times with PBS/0.5% BSA and pelleted at 4000 rpm for 3 min, and the pellet was resuspended in 1 mL PBS/BSA. Antibody-stained EV-coated beads were analyzed on a flow cytometer (FACSCalibur, BD Biosciencs, Franklin Lakes, NJ, USA) and fluorescence compared with specific antibodies and relevant isotype controls (Mouse IgG1K, eBioscience).

### 2.8. Western Blot Analysis

Western blot was performed as described previously [57]. Briefly, cell lysates or isolated EV samples (containing 10 μg protein) were resolved via 12.5% SDS-PAGE, transferred to PVDF membrane, and probed using anti-CD9 (catalog number: 10626D, Invitrogen) and anti-TSG 101 (catalog number: sc-7964, Santa Cruz Biotechnology). Subsequently, blots were incubated with horseradish peroxidase (HRP) conjugated secondary antibody (1:3000) for 1 h at room temperature. Washing steps after antibody incubations were conducted on an orbital shaker four times at 10 min intervals with TTBS. Blots were developed with chemiluminescent reagent (Pierce, Waltham, MA, USA). Original blots see Appendix A.

### 2.9. In-Solution Digestion of EVs for 2D-LC-MS/MS

Isolated EVs were buffer-exchanged into 50 mM ammonium bicarbonate and prepared for analysis of protein profiles by reducing with 10 mM dithiothreitol (DTT; Merck, Darmstadt, Germany) at 56 °C for 1 h and alkylating with 30 mM iodoacetamide (IAA, Sigma, St. Louis, MO, USA) in the dark for 30 min at room temperature. After removing excess alkylating agent with 10 mM DTT at 56 °C for 10 min, protein mixtures (50 μg) were digested with 1 μg of sequencing-grade modified porcine trypsin (Promega, Madison, WI, USA) at 37 °C for 18 h. Peptide mixtures from the tryptic digestion of 50 μg EV proteins were reconstituted in 0.1% formic acid (FA; Sigma-Aldrich, Saint Louis, MO, USA), desalted using a homemade microcolumn (Source 15RPC; GE Healthcare), and analyzed via 2D HPLC coupled with a linear ion trap mass spectrometer (LTQ Orbitrap MS; Thermo Fisher, San Jose, CA, USA) operated with Xcalibur 2.2 software (Thermo Fisher).

### 2.10. Two-Dimensional LC-MS/MS Analysis (2D-LC-MS/MS) 

Dried peptides (30 µg) were reconstituted in 50 µL HPLC buffer A (30% acetonitrile/0.1% formic acid) and loaded onto a homemade strong cation exchange (SCX) chromatography column (Luna SCX 5 µm, 100 Å, 0.5 × 255 mm; Phenomenex, Torrance, CA, USA) at a flow rate of 5 µL/min for 30 min. Peptides were eluted using a gradient of 0–100% HPLC buffer B (0.5 M ammonium chloride/30% acetonitrile (ACN)/0.1% FA) and separated into 44 fractions using on-line two-dimensional high-performance liquid chromatography (2D-HPLC; Dionex Ultimate 3000, Thermo Fisher). Each SCX fraction was further diluted in-line prior to the trap of the reverse-phase column (Zorbax 300SB-C18 5 µm, 300 Å, 0.3 × 5 mm; Agilent Technologies, Wilmington, DE, USA) and diluted peptides separated on a homemade column (Synergi Hydro-RP 2.5 µm, 100 Å, 0.075 × 200 mm with a 15 μm tip; Phenomenex, Torrance, CA, USA). A linear gradient of fractionation was applied as follows: 3–28% HPLC buffer C (99.9% ACN/0.1% FA) for 37 min, 28–50% buffer C for 12 min, 50–95% buffer C for 2 min, 95% buffer C for 5 min, and 3% buffer C for 9 min at a flow rate of 0.3 μL/min. The LC apparatus was coupled with a 2D linear ion trap mass spectrometer (LTQ-Orbitrap ELITE; Thermo Fisher) operated using Xcalibur 2.2 software (Thermo Fisher). Full-scan MS was performed in Orbitrap over a range of 400–2000 Da and resolution of 60,000 at m/z 400. Internal calibration was performed using the ion signal of [Si (CH_3_)_2_O]_6_H^+^ at m/z 445.120025, 462.146574, and 536.165365 as lock masses. A total of 12 data-dependent MS/MS scan events (12 CID) were followed by 1 MS scan for the 12 most abundant precursor ions in the preview MS scan.

### 2.11. Data Processing

The raw files of resulting MS/MS spectra obtained from LTQ-Orbitrap MS were searched against 21,219 entries of Homo sapiens in the SwissProt-human database released in 20150429 using Proteome Discoverer 1.4 (Thermo Fisher). The cleaved enzyme was set to “trypsin”, with a maximum of one missed cleavage site. The precursor mass tolerance was set to 10 ppm, and fragment ions mass tolerance to 0.6 Da. Fixed modification was set to “carbamidomethylation at cysteine” and variable modifications to “acetylation at protein N-term “, “oxidation at methionine” and “deamidation at N-term glutamine”. Following the MASCOT search, the score thresholds of protein and peptide were 20 and 15, respectively. The peptide identification confidence threshold was set to “1% false discovery rate (FDR)” in the workflow of Proteome Discoverer 1.4. Peptide Validator algorithm was applied in calculation of FDR for peptide sequence analysis to distinguish true positives from random matches (decoy database).

### 2.12. Gene Ontology Annotation and Topological Analysis of Target Proteins

To determine the subcellular localization and cellular compartments of the proteins involved, Human Protein Reference Database (HPRD) (http://www.hprd.org/), Protein ANalysis THrough Evolutionary Relationships (PANTHER) consortium databases (http://www.pantherdb.org/) (accessed on 8 June 2016) and Fun Rich Functional Enrichment Analysis Tool (http://www.funrich.org/) (accessed on 20 June 2016) were searched. Topological analysis of target proteins was performed using Protter (http://wlab.ethz.ch/protter/start/) (accessed on 15 September 2018), an open web application that integrates protein sequence features and transmembrane topology into illustrations [58].

### 2.13. Selection of Signature Peptides and Accurate Inclusion Mass Screening (AIMS)

Target proteins were subjected to in-silico tryptic digestion using MS digest software (https://prospector.ucsf.edu/prospector/html/instruct/digestman.htm) (accessed on 10 June 2022), and the signature peptides for each candidate protein selected based on the following criteria: (a) unique peptides containing 8–20 residues without known post-translational modification sites (determined from HPRD and no sequential or missed trypsin cleavage sites; (b) peptides without chemically reactive amino acids, such as C, M, and W; (c) peptides without unstable sequences, such as NG, DG, and QG; and (d) peptides without sequences potentially leading to missed cleavage, such as RP and KP. To perform AIMS, peptide samples were reconstituted in HPLC buffer A (0.1% FA), loaded across a trap column (Zorbax 300SB-C18, 0.3 × 5 mm; Agilent Technologies) at a flow rate of 0.25 μL/min in HPLC buffer A, and separated on an analytical C18 column (Synergi Hydro-RP 2.5 µm, 0.075 × 200 mm with a 15 μm tip; Phenomenex, Torrance, CA, USA). Eluted peptides were analyzed using a two-dimensional linear ion trap mass spectrometer LTQ-Orbitrap. Intact peptides were detected in the Orbitrap at a resolution of 60,000. Internal calibration was performed using the ion signal of (Si (CH_3_)_2_O)_6_H^+^ at *m*/*z* 445.120025 as a lock mass. Accurate *m*/*z* values were initially inputted into the inclusion list, and MS/MS analysis performed on the LTQ–Orbitrap MS with a 10 ppm parent mass tolerance to validate peptide detection. 

### 2.14. EV Digestion and iTRAQ Labelling of EV Derived Peptides

EVs were isolated from plasma samples pooled from healthy controls and CRC cases using ultracentrifugation as described in Section 2.4. To EV samples (containing 50 µg protein), 1 M triethylammonium bicarbonate buffer (TEABC) was added at a final concentration of 100 mM. Next, samples were treated with 250 mM tris(2-carboxyethyl) phosphine (TCEP) to a final concentration of 5 mM. Reduced samples were incubated at 60 °C for 30 min, alkylated with IAA (10 mM), and incubated in the dark for 30 min at room temperature. Subsequently, 100 mM TEABC was added, followed by digestion with 5 µg trypsin at 37 °C for 16 h. The reaction was terminated with 1.5% FA and 0.3% trifluoroacetic acid (TFA). Digested samples were desalted using C18 beads (15 µL; LiChroprep RP-18, Merck, Darmstadt, Germany) and labeled with iTRAQ reagents (114 for HC and 116 for CRC samples) at room temperature for 2 h. Samples were desalted with C18 beads (15 µL).

### 2.15. 2D LC-MS/MS Analysis Coupled with iTRAQ

Tryptic peptides (36 µg) were reconstituted in HPLC mobile phase A (30% ACN/0.1% FA) and loaded onto a homemade SCX column (Luna 5 µm SCX, 0.5 × 255 mm; Phenomenex). Peptides were eluted over a gradient of 0–100 % HPLC mobile phase B (1 M ammonium nitrate/25% ACN/0.1% FA) and separated into 72 fractions using on-line 2D-HPLC (Dionex Ultimate 3000; Thermo Fisher). Each SCX fraction was further 40-fold diluted in-line prior to the trap of reverse-phase column (Zorbax 300SB-C18 5 µm, 0.3 × 5 mm; Agilent Technologies). Diluted peptides were resolved on an analytical C18 column (Synergi Hydro-RP 2.5 µm, 0.075 × 200 mm with a 15 μm tip; Phenomenex). Linear gradient of fractionation was applied as follows: 6–9% HPLC mobile phase C (99.9% ACN/0.1% FA) for 2 min, 9–26% mobile phase C for 28 min, 26–35% mobile phase C for 10 min, 35–45% mobile phase C for 5 min, 45–63% mobile phase C for 4 min, 63–95% buffer C for 2 min, 95% buffer C for 3 min, and 6% buffer C for 5 min at a flow rate of 0.25 μL/min. The LC apparatus was coupled with a two-dimensional linear ion trap mass spectrometer LTQ-Orbitrap ELITE. Full-scan MS was performed on Orbitrap over a range of 400–2000 Da and resolution of 60,000 at *m*/*z* 400. Internal calibration was performed using the ion signal of [Si (CH3)2O]_6_H+ at m/z 445.120025, 462.146574, and 536.165365 as lock masses. The 12 data-dependent MS/MS scan events (6 collision-induced dissociation (CID) and 6 high energy collision-induced dissociation (HCD) modes) were followed by 1 MS scan for the 6 most abundant precursor ions in a preview MS scan.

### 2.16. Sequence Database Search and Quantitative Data Analysis for iTRAQ

Data analysis was performed using Proteome Discoverer software (version 1.4, Thermo Fisher Scientific) involving the reporter ion quantifier node for iTRAQ quantification. MS/MS spectra were searched against the Swiss-Prot human sequence database (released on 20190213, selected for *Homo sapiens*, 20418 entries) using the Mascot search engine (Matrix Science, London, UK; version 2.5). For protein identification, the precursor mass tolerance was set to 10 ppm and fragment ion mass tolerance to 0.6 Da for the CID mode via ion trap analysis and 0.05 Da for the HCD mode by Orbitap analysis, with allowance for one missed cleavage from tryptic digestion. Fixed modification was set to methylthiolation at cysteine (+45.99 Da) and variable modification to acetylation at the protein N-terminus (+42.01 Da), oxidation at methionine (+15.99 Da), pyroglutamate conversion at N-terminal glutamine (−17.03 Da), and iTRAQ 4-plex labeling at lysine and peptide N-terminus (+144.10 Da). Based on Mascot search results, the score threshold for peptide identification was set to “1% false discovery rate (FDR)” in the processing workflow and Peptide Validator algorithm applied in calculation of FDR for peptide sequence analysis to distinguish true positives from random matches (decoy database). The decoy database was generated with Mascot with a similar size, including number of amino acids and proteins, as the original normal database. In iTRAQ quantification, each reporter ion was integrated by the mode of most confident centroid at 20 ppm tolerance, and iTRAQ-114 (114.11 Da) set as the denominator and iTRAQ-116 (116.11 Da) as the numerator for generation of a quantifiable ratio. Proteins with one unique peptide hit or without a quantifiable ratio were removed. Proteins with ratios above the mean plus one standard deviation (SD) of all ratios were classified as upregulated and those with ratios below the mean minus one SD as downregulated.

### 2.17. Synthesis of Surrogate Peptides

Internal standard peptides coded with stable isotope (SIS peptides) by labeling ^13^C/^15^N of Lys and Arg, leading to a 8 and 10 Da increase in the weight of peptides harboring Lys and Arg, respectively, were synthesized and purchased from Thermo Fisher Scientific. The purity of SIS peptides was >95% (in the majority of cases, purity was >98%), as determined via mass spectrometry analysis after HPLC purification. Synthetic light peptides (without stable isotope labeling) were purchased from Kelowna International Scientific (Taipei, Taiwan).

### 2.18. Preparation of EV Samples for MS-Based Targeted Protein Quantification 

EVs were isolated as described earlier. EVs (5 μg) were dissolved in PBS containing 100 mM Tris buffer at pH 8.5. Samples were reduced with 5 mM TCEP at 60 °C for 30 min and alkylated with 10 mM IAA in the dark for 30 min at room temperature. After 2-fold dilution with 100 mM Tris, samples were digested with 0.2 μg trypsin at 37 °C for 16 h. A standard cocktail of 14 SIS peptides (containing 50 fmol ^13^C/^15^N-labeled peptides at Lys or Arg) was suspended in each EV sample followed by treatment with 0.5% FA and 0.2% TFA to acidify samples and terminate tryptic digestion. The digested samples were desalted with solid-phase extraction Oasis HLB (30 μm) cartridges (Waters, Milford, MA, USA) and lyophilized for further quantitative MS analysis using product ion scanning (PIS) mode.

### 2.19. LC-PIS-MS, MS Data Processing and Generation of Response Curves

Dried peptides (1 µg) were reconstituted in HPLC mobile phase A (0.1% FA), loaded across a trap column (Zorbax 300SB-C18 5 µm, 0.3 × 5 mm; Agilent Technologies) at a flow rate of 20 μL/min in HPLC mobile phase A, and resolved on an analytical column (ACQUITY UPLC C18 1.7 μm, 0.1 × 100 mm) using HPLC mobile phase B (100% ACN/0.1% FA) at a flow rate of 0.4 µL/min. A linear gradient of 8–35% HPLC mobile phase B was applied in separation of the 14 target peptides for 11.3 min and a two-step 95% HPLC mobile phase B subsequently applied in regeneration to avoid carry-over into the next sample run. Eluted peptides were analyzed with a two-dimensional linear ion trap mass spectrometer (LTQ-Orbitrap ELITE). Intact peptides were detected in the Orbitrap at a resolution of 120,000. Internal calibration was performed using the ion signal of (Si (CH_3_)_2_O)_6_H^+^ at *m*/*z* 536.165365 as a lock mass. Accurate m/z values of precursor ions and their retention times were initially imported into the inclusion list. MS/MS analysis was subsequently performed on each targeted precursor and the generated product ions detected in the linear trap quadrupole (LTQ) following the scheduled scan times. The automatic gain control (AGC) value of MS and MS/MS were set to 3 × 10^6^ ions (full scan) at 1000 milliseconds (ms) and 5 × 10^3^ (CID) at 300 ms for maximum accumulated time or ions, respectively.

RAW files of the spectra obtained from LTQ-Orbitrap ELITE were searched against peptide sequences derived from the 14 target proteins. The SEQUEST algorithm was used for data processing and search results integrated using Proteome Discoverer 1.4 (Thermo Fisher). The MS tolerance for the monoisotopic peptide window was set to 10 ppm, and MS/MS tolerance set to 0.6 Da. Dynamic modification was set with stable isotope-containing lysine (+8 Da) and arginine (+10 Da), and charge states of the peptides set to 2+ and 3+. The false discovery rate (FDR) calculated using peptide sequence analysis to distinguish true positives from random matches (decoy database) was set to 1% as the cut-off threshold for ensuring confidence of peptide identification. Spectral libraries of sample runs constructed from PD software in msf format and relevant RAW files were imported into Skyline software (v. 21.1.0.278). Product ions with the 10 highest values of intensity were automatically selected with Skyline from ion 1 to the last ion with 1+, 2+ and 3+ ion charges in the 300–1250 *m*/*z* range and the ion match tolerance set to 0.7 *m*/*z*. After manually removing ions with interference and checking their position of retention time, the integral area of a trapezoidal model with an unsmoothed chromatogram was applied to determine the final peak area. The specific peaks of endogenous peptides were identified according to co-elution with exogenously supplemented SIS peptides and the peak areas of 2 to 7 selected fragments of target peptides summarized for further quantitative analysis. The concentration of the target peptide in samples was measured based on the ratio of the peak area to known concentrations of SIS peptides. Three independent technical repeats were performed (from digestion to the final LC-PIS-MS step) for each plasma EV sample.

Response curves for the 14 peptides were generated from experiments conducted in quintuplicate. For each target, serially diluted heavy peptide (0, 0.049, 0.098, 0.195, 0.391, 0.781, 1.563, 3.125, 6.25, 12.5, 25, 50, and 100 fmol) and a constant amount of light peptide (10 fmol) were spiked into an EV-protein digest background (1 µg protein) and analyzed via LC-PIS-MS. The MRM statistical software, QuaSAR was applied to assess the limit of detection (LOD) using the “blank and low concentration sample” method to estimate the mean of the blank samples and standard deviations of blank and low-concentration samples [59]. The lower limit of quantification (LLOQ) was calculated as the LOD value multiplied by 3 [60].

### 2.20. Statistical Analysis

Bar graphs for iTRAQ were generated using Sigma plot software. Data are presented as means ± standard deviation. The Mann–Whitney test was used to compare the differences in plasma derived EV levels of selected proteins between HC and CRC groups. Mann–Whitney and Kruskal–Wallis tests were employed to evaluate the association of plasma-EV ADAM10, CD59, TSPAN9 and CEA with various clinicopathological parameters of CRC patients. The diagnostic power of ADAM10, CD59, TSPAN9 and CEA was analyzed by constructing a receiver operating characteristic (ROC) curve with sensitivity versus 1-specificity and calculating the area under the ROC curve (AUC). For all statistical analyses, a two-tailed *p*-value ≤ 0.05 was considered significant. Calculations and diagrams were generated using GraphPad Prism 7.0 (GraphPad Software, Inc., San Diego, CA, USA).

## 3. Results

### 3.1. Study Design and Characterization of EVs

To find potential biomarker candidates, we isolated EVs from the conditioned media of four different CRC cell lines and utilized 2D-LC-MS/MS to identify total EV proteins. Bioinformatics analyses of the identified proteins were applied to reveal EV membrane proteins expressed in common among at least three CRC cell lines, as well as their membrane topology. The presence of these membrane protein targets in clinical plasma EV specimens was evaluated using an AIMS analysis of pooled plasma EV samples from 30 CRC patients and 30 HCs (dataset 1). An iTRAQ-based proteomics approach was further used to quantitatively compare the pooled plasma EV protein profiles between these CRC patients and HCs (dataset 2). A candidate biomarker list was then generated by integrated analysis of the 2 datasets, and tier 1 candidates containing 13 EV membrane proteins and the single EV marker, CD9, were selected for verification by targeted MS using product ion scanning (PIS) mode in 153 individual plasma EV samples isolated from 73 CRC patients and 80 HCs. The workflow of our study design is summarized in Figure 1.

EVs were purified from the conditioned media of HT29, SW480, Colo205 and SW620 cells by ultracentrifugation, and the quality and purity of preparations were evaluated using several types of analyses prior to application of proteomics workflows. The presence of the marker, CD9, on the EV surface was examined by flow cytometry after coupling of EVs to latex beads. The principle of this FACS analysis is highlighted in Figure 2A. CD9, a tetraspanin protein, was found to be highly expressed in EV fractions prepared from all four CRC cell lines studied (Figure 2B). Electron microscopy revealed nanovesicular entities in these preparations within a size range consistent with their characterization as small EVs (30–150 nm) (Figure 2C). An examination of the size distribution of EVs in culture supernatants from these CRC cell lines using a NanoSight particle tracking analysis revealed a wide range of mean EV diameters across cell lines (HT29, 112 nm; SW480, 150 nm; SW620, 156 nm; Colo205, 120 nm) (Figure 2D). To investigate the expression of expected reported EV markers as well as the relative expression of these markers compared with the parent cell as a whole, we performed Western blot analyses comparing whole-cell extracts with EVs (10 μg each). These analyses showed substantial enrichment of the multivesicular body marker TSG101 (tumor susceptibility 101) and cell surface marker CD9 (Figure 2E).

### 3.2. Identification of EV Proteins by 2D-LC-MS/MS and Annotation of Identified Proteins

For identification of proteins in EVs generated by the four CRC cell lines, peptides recovered from in-solution trypsin digestion of EVs were subjected to two-dimensional liquid chromatography tandem mass spectrometry (2D-LC-MS/MS). We were able to identify more than 2100 EV proteins from high-confidence peptide sequences in each of the four cell lines with an error rate of less than 1% (Appendix A), including most of the proteins listed in the EV databases, Exocarta and Vesiclepedia, and identified in earlier EV protein research. As expected, tetraspanins, annexins, integrins, and a host of other membrane and cell surface proteins were found, including the conventional EV markers, CD9, CD63, CD81, TSG101, ALIX, HSPA8, and HSP90AB1.

The Exocarta-derived and Gene Ontology (GO)-based functional enrichment tools FunRich and Panther, respectively, were used to classify the identified EV proteins by subcellular location and sort the MS-identified proteins into their appropriate compartments (Figure 3A). Given the focus on membrane proteins, all membrane proteins, including cell surface, transmembrane, integral and peripheral proteins, were aggregated into a single category, resulting in a total of 754, 666, 503 and 780 membrane proteins in HT29, SW480, SW620 and Colo205 cells, respectively (Figure 3B). A total of 518 target membrane proteins commonly identified across at least 3 cell lines (332 for all 4 cell lines and 186 for three cell lines) were chosen for further verification after a Venn diagram analysis of membrane proteins in common (Figure 3C). The list of these 518 membrane proteins and their cell compartment annotations are shown in Appendix A. A topological analysis revealed that more than half of these 518 target proteins were single- or multiple-transmembrane domain-containing proteins or anchored proteins (Appendix A). Figure 3D shows MS-detected peptide(s) for selected examples of topological analysis of target membrane proteins, including neutral amino acid transporter B (containing multiple-transmembrane domains), integrin beta-3 (containing a single transmembrane domain) and CD59 (containing a lipid anchor). 

### 3.3. Prioritization of Target Proteins and Confirmation of Their Presence in Plasma-Derived EVs from HCs and CRC Patients

After identification of EV membrane proteins in common among CRC cell lines, we next applied LC-MS/MS running in AIMS mode to detect the presence of candidate proteins in plasma-derived EVs from HCs and CRC patients. Only those proteins that are found by AIMS, which is used to determine whether candidate proteins are detectable in plasma samples, are sent to a second verification stage [51]. To perform AIMS, we first generated a list of tryptic peptides from the 518 membrane protein targets in silico and then inspected these peptides using various filters, as described in Materials and Methods, to select signature peptides (1–4 peptides) for each candidate protein, resulting in a total of ~1400 peptide targets. To this end, we isolated EVs from plasma samples pooled from HCs and CRC cases (n = 30 for each group). The mean size of EVs isolated from pooled controls and patients was determined to be 107.5 nm and 97.1 nm, respectively (Figure 4A). The presence of CD9 in isolated EV preparations was validated by flow cytometry (Figure 4B). Thereafter, EVs were subjected to tryptic digests and AIMS analysis. Among the 518 membrane protein targets, we were able to identify a total of 107 target proteins, including 24 solely in the CRC group, 62 in both CRC and HC groups, and 21 solely in the HC group (Figure 4C and Appendix A). AIMS-detected target proteins exclusive to CRC cases (n = 24) and those discovered in both HCs and CRC patients (n = 62) were chosen for further analysis. 

The 24 target proteins detected only in CRC cases were given primary consideration, and were ranked in descending order based on their peak area detected by MS. The 62 target proteins detected in both CRC patients and HCs were given second priority. We then calculated the fold-change of these proteins between CRC and HC groups, estimated based on peak area detected by MS, to determine whether they were up- or down-regulated in the CRC group. This analysis revealed 36 up-regulated proteins in the CRC group 12 of which showed a fold-change greater than 2.5. These 36 target proteins (24 + 12) were chosen for further investigation (Appendix A). 

### 3.4. Quantitative Proteome Profiling of EV Samples from CRC Patients and HCs

We next applied the iTRAQ-based proteomics approach to quantitatively compare the proteome profiles of the isolated EV samples used for the aforementioned AIMS experiment. This allowed us to confirm whether we could detect the AIMS-detected candidates and additionally identify proteins that were differentially expressed between CRC and HC groups. Starting from 50 μg protein, we quantified a total of 1143 EV proteins. Among the quantified proteins, the levels of 200 proteins were elevated (>mean +1 standard deviation [SD]) and those of 198 proteins were reduced (<mean −1 SD) in EV samples of CRC patients (Figure 4D). A complete list of quantified proteins in EV samples from CRC patients and HCs with up- and down-regulation information is presented in Appendix A. Furthermore, we were able to detect 87 of the 107 proteins found by AIMS, as well as 72 proteins from the 518-protein target list that AIMS was unable to detect (Figure 4E and Appendix A). 

### 3.5. Selection of Candidates for MS-Based Targeted Protein Quantification

On the basis of AIMS and iTRAQ results described above, we selected 58 potential targets for verification according to the fold-change between HC and CRC groups as well their expression levels in CRC cases (Figure 4F). We then further categorized the prioritized candidates into 3 groups: the first (tier 1) included all membrane proteins (cell surface, peripheral and integral proteins; n = 25); the second (tier 2) included membrane-associated proteins (n = 20); and the last (tier 3) comprised unclassified proteins (n = 13) (Appendix A). To learn more about these 58 proteins, we also performed a literature search, which revealed that some of these proteins were previously reported to be up-regulated in CRC tissues and serum in small and medium cohorts by various research groups. However, their overexpression or down-regulation in plasma-derived EVs had not been investigated (Appendix A). 

Premised on the assumption that surface proteins are readily accessible in plasma EVs, we selected only tier 1 proteins for further verification. Selected proteins with one peptide each were scrutinized for any modifications, as previously noted and then a BLAST search was conducted to determine whether these peptides were unique to the target protein. We then again checked the expression of these 25 tier 1 proteins by AIMS in EVs isolated from individual plasma samples from HCs (n = 5) and CRC patients (n = 6), selecting a final set of 13 proteins based on the ratio between CRC and HC groups, using both fold-change greater than 1.5 and more than one detectable case in CRC as criteria for further verification by MS-based, targeted protein quantification. The AIMS results for these 25 tier 1 proteins are shown in Appendix A. The 13 potential targets and CD9, a well-known EV marker, together with their peptide sequence, are shown in Table 1.

### 3.6. Quantification of Selected Targets in Individual Plasma-Derived EVs by Targeted MS 

To quantify the selected targets in plasma derived EVs, we first synthesized 14 surrogate (light and heavy) peptides representing the 14 selected proteins and developed a multiplex LC-MS/MS assay run in PIS mode to measure the 14 targets, as shown schematically in Appendix A. We constructed reverse response curves for the 14 target peptides using a scheduled PIS assay in LTQ-Orbitrap ELITE to assess the performance of this newly developed targeted MS assay. The response of a serially diluted heavy peptide (0.488–100 fmol, 12 data points) mixed with a constant amount of light peptide (10 fmol) for each target was analyzed in a background of plasma EV digest (1 μg protein). The response curves of all 14 target peptides are shown in Appendix A, and the linearity of the response curves and values for limit of detection (LOD), lower limit of quantification (LLOQ), and coefficient of variation (CV) for all 14 targets are detailed in Appendix A. Most of these 14 targets (13/14, 93%) were detected in more than 9 points on the 12-point response curve (blank plus 12 points of 2-fold serial dilutions) with a good linear response (R^2^ > 0.97). A total of 11 targets showed LLOQ values for peptides less than 1 fmol/μg, whereas 3 targets (ALCAM, CD9 and TSPAN9) had higher LLOQ values ranging from 1.072 to 1.943 fmol/μg. The LLOQ value for each target is also expressed as protein concentration (in ng/mL), which represents the amount of protein corresponding to the determined level of proteotypic peptide, assuming complete tryptic digestion (i.e., 100% recovery), based on total protein concentration (0.0387 μg/μL) of the background EV protein digest. Eight targets (ADAM10, APMAP, ART4, CD58, CD59, ICAM3, RHAG and TSPAN33) had LLOQ values less than 1 ng/mL protein. The remaining six targets (ALCAM, CD9, ITGAM, SELP, TSPAN9 and TTYH3) had LLOQ values between 1 and 5 ng/mL. The median CV value, calculated in the linear range of the response curve for each target, was less than 12% for all 14 targets, indicating high stability in the assay (Appendix A). Collectively, these data indicate that the newly developed multiplexed assay shows good stability, linearity, and target-dependent LLOQ in plasma EV samples containing 1 μg protein, with a majority of targets exhibiting LLOQ values less than 1 fmol/μg (or 2 ng/mL).

We then applied this multiplexed assay to quantify the selected targets and CD9 in individual plasma-derived EVs collected from 80 HCs and 73 CRC patients. These quantitative results are summarized in Table 2. The median values of CV for each target were calculated in triplicate experiments in these 153 samples. Although four targets (ALCAM, RHAG, TSPAN33 and TTYH3) had a median CV greater than 20%, the results for the other nine targets and CD9 illustrate the superior precision of this assay for use on plasma-derived EVs (Appendix A). Results of quantification summarized in Table 2 demonstrate that six targets (ADAM10, ALCAM, APMAP, CD58, CD59 and TSPAN9) and CD9 could be quantified in more than 50% (37 cases) of CRC samples, but two targets (ART4 and SELP) were detectable in less than 25% (18 cases) of CRC samples. The quantifiable range of these 13 targets and CD9 ranged from 0.003 ng/mL (for ART4) to 5.50 ng/mL (for CD9) in HCs and from 0.13 ng/mL (for ART4) to 7.66 ng/mL (for CD9) in CRC patients. Notably, the average concentrations of most of the 14 targets in the CRC group were higher than their respective LLOQ values; the only exceptions were ALCAM and SELP (Appendix A). These observations support the feasibility of our newly developed targeted MS assay to quantify the 14 selected targets in plasma-derived EVs from CRC patients. A comparison of the levels of the 13 targets between HCs and CRC patients showed that 10 targets (ADAM10, ALCAM, APMAP, ART4, CD58, CD59, ICAM3, ITGAM, RHAG and TSPAN9) were significantly (*p* < 0.05) upregulated (1.72- to 50.49-fold) in CRC patients. Among them, three targets, ADAM10, CD59 and TSPAN9, with AUC values of 0.83, 0.95 and 0.87, respectively, showed high-level ability to distinguish CRC patients and were selected for further analysis (Table 2 and Figure 5A). 

We further compared the levels of these three targets in early-stage CRC patients (TNM stage I and II, n = 32) and HCs. Both fold changes (CRC/HC = 2.66, 4.16 and 1.93 for CD59, ADAM10 and TSPAN9, respectively) and AUC values (0.85, 0.96 and 0.86 for ADAM10, CD59 and TSPAN9, respectively) represented significant increases, indicating high diagnostic ability in early-stage CRC patients (Appendix A). Notably, both CD9 levels and total EV protein concentrations were nearly equal among HCs, all CRC sample, and early-stage CRC specimens, indicating that fold-changes of the 13 targets were not influenced by the amount of EVs (Table 2 and Appendix A, and Figure 5B,C). We also examined the plasma levels of CEA in the same samples. This analysis showed that only a small fraction of CRC patients had high CEA levels and that there was no significant difference in CEA levels between HC and CRC groups (Table 2 and Figure 5D). Collectively, these data indicate that ADAM10, CD59 and TSPAN9 may be good plasma EV biomarkers for CRC.

### 3.7. Generation of Candidate Plasma EV Protein Biomarker Panels 

To establish a biomarker panel of plasma-derived EVs for CRC detection, we used logistic regression to process the quantitative results acquired for the 5 targets (ADAM10, APMAP, CD58, CD59 and TSPAN9) that could be quantified in more than 50% (37 cases) of CRC samples with average concentrations higher than their respective LLOQ values in 153 subjects from HC (n = 80) and CRC (n = 73) groups. Two proteins, CD59 and TSPAN9, were ultimately selected as a biomarker panel by the analysis, and an ROC curve was generated from values of measured probability (Figure 6A,C). The area under the curve was calculated to be 0.98, and the corresponding sensitivity and specificity were calculated to be 98.6% and 91.3%, respectively, based on the risk score (0.211) obtained from the best cutoff point for distinguishing HC and CRC groups (Figure 6D). Using the same algorithm and risk score, we also calculated the probability of the early-stage CRC group compared with the HC group. As shown in Figure 6B,E, the risk score (0.211) displayed superior diagnostic power to discriminate the early-stage CRC group from the HC group, as evidenced by the AUC value of 0.99, 100% sensitivity, and 96.3% specificity. 

### 3.8. Association of Plasma-Derived EV Levels of ADAM10, CD59 and TSPAN9 and Plasma CEA Levels with Clinicopathological Characteristics of CRC Patients

Finally, we explored the potential association of plasma-derived EV levels of ADAM10, CD59, TSPAN9 and plasma CEA with different clinicopathological characteristics, including gender, age, TNM stage, tumor stage, lymph node metastasis and distant metastasis, of enrolled CRC patients (Table 3). This analysis showed that (i) none of the measured values for the four proteins (ADAM10, CD59, TSPAN9 and CEA) were significantly associated with gender or age; (ii) higher CD59 protein levels were significantly correlated with distant metastasis (*p* = 0.0475); (iii) higher TSPAN9 protein levels were significantly correlated with lymph node metastasis (p = 0.0011), distant metastasis (*p* = 0.0104) and higher TNM stage (*p* = 0.0065); (iv) higher plasma levels of CEA protein were significantly correlated with higher tumor stage (*p* = 0.0035), lymph node metastasis (*p* = 0.0003), distant metastasis at diagnosis (*p* = 0.0252) and higher TNM stage (*p* = 0.0010); and (v) protein levels of ADAM10 did not show a significant correlation with clinical characteristics (Table 3).

## 4. Discussion

It is well recognized that EVs carry many membrane proteins on their surface that can transduce signals to recipient cells and serve as biomarkers and target proteins for the diagnosis and treatment of diseases [61,62]. A number of EV membrane proteins, including CD147/basigin and EpCAM, have been selected to pair with the EV markers CD9 or CD63 as biomarker panels for CRC and have been successfully tested for effective cancer detection using blood specimens [40,41]. These promising findings prompted the present effort to discover novel CRC plasma EV biomarkers, with a focus on membrane proteins. Through in-depth analysis of EV membrane protein profiles from CRC cell lines and an iTRAQ quantitative analysis of differentially expressed plasma EV proteins from CRC patients and HCs, we generated a biomarker candidate list. We further prioritized the candidate targets by AIMS-based detection of all possible candidates in plasma EVs from CRC patients and HCs, and successfully verified the prioritized candidate targets by targeted MS-based absolute quantitative analysis of prioritized candidates in individual plasma EV samples from age/sex-matched CRC patients and HCs. Finally, we generated a novel two-marker panel consisting of EV membrane proteins CD59 and TSPAN9 that effectively discriminated CRC patients from HCs (AUC = 0.98). To our knowledge, this is the first study to apply targeted MS for verification of multiple (in our case, more than a dozen) EV membrane proteins as plasma cancer biomarkers in which the amounts of target proteins were quantitatively measured in individual plasma EV samples by spiked SIS peptides with known quantities. Thus, for the first time, we were able to simultaneously measure the absolute levels of multiple EV membrane proteins in individual plasma EV samples and perform head-to-head comparisons of selected targets’ ability to detect CRC from the same clinical sample set. Such a targeted MS-based biomarker verification pipeline has been successfully applied to identify promising body fluid-accessible biomarkers or biomarker panels for detection of cancer and other diseases [63,64,65,66,67]. Candidate proteins that stand out from the remaining targets in this kind of head-to-head comparison have a better chance of surviving and becoming clinically applicable biomarkers during further validation testing using samples from a large cohort. For example, we recently validated saliva matrix metalloproteinase 1 (MMP-1) as a strong diagnostic marker of oral cavity cancer in saliva samples from a large cohort (n = 1160), supporting previous reports that this protein is one of the most promising salivary biomarkers for oral cavity cancer using a targeted MS-based assay to compare multiple candidate proteins [66,68].

Advances in MS-based proteomics methods have greatly enhanced the biomarker discovery and verification pipeline through targeted and non-targeted approaches. Proteome cataloguing of EVs from various cancer types has revealed numerous membrane and cytosolic proteins, as well as a set of proteins distinct for different types of malignancies, that reflect the original host cell. For the most part, discovery efforts have focused on tissues, proximal fluids or cell lines, with plasma being used only after candidates have been found. However, there are no reliable principles for determining which tissue or cellular proteins will work as plasma biomarkers, and the error rate is considerable. In general, clinical data are rarely used in discovery attempts to select markers that will have the best chance of success in the targeted clinical environment. As a result, discovery efforts generate vast lists of candidate biomarkers, but they also allow for the testing of the greatest number of biomarkers possible, increasing the chances of finding clinically meaningful indicators. 

AIMS, implemented before committing to the time and resource-intensive procedures of developing a quantitative test, allows for prioritizing of a large number of biomarker candidates based on their detection in plasma. It is a remarkable technique for bridging the gap between unbiased discovery and MS-based focused assay creation [51]. Following this approach, we employed AIMS as an empirical prioritizing step to determine whether biomarker candidates could be detected in plasma and additionally used iTRAQ to validate the targets detected by AIMS (Figure 4). Among the 13 prioritized targets, eight (ADAM10, ALCAM, CD58, CD59, ITGAM, SELP, TSPAN9 and TTYH3) were previously reported to be dysregulated in tissue and/or plasma specimens from CRC patients [69,70,71,72,73,74,75,76]. Even though many of the selected candidates had been reported as potential markers for CRC, only three targets (ADAM10, CD59 and TSPAN9) showed good power (AUC > 0.8) to discriminate CRC patients from HCs in the current study (Table 2 and Appendix A). Our findings echo previous observations regarding the high error rate for verification of liquid biopsy biomarkers and support the need for further clinical validation of verified biomarkers/biomarker panels. 

CD59, a GPI-anchored membrane protein, inhibits the development of the membrane attack complex, preventing complement activation and perhaps protecting cancer cells from complement-dependent cytotoxicity [77,78]. Recent studies have indicated that CD59 is highly expressed in several cancer cell lines and tumor tissues, including those from CRC patients [78]. It has been shown that expression levels of membrane-bound complement regulatory proteins (mCRPs), including CD46, CD55 and CD59, are considerably higher in colon cancer tissues than in normal neighboring normal colon tissues. Furthermore, an analysis based on TNM staging showed that the expression of CD55 and CD59 is higher in stage III and stage IV colon cancers than in stage I and stage II tumors [79]. Another study that applied tissue microarrays to evaluate the immunohistochemical expression of CD59 in over 460 well characterized CRC cases similarly found an association between high CD59 expression and worse prognosis, a relationship that held particularly well for patients in the early stages of the disease [71]. CD59 has also been reported to be overexpressed in several other cancers, including breast, esophageal and pancreatic cancer, and that its expression may facilitate tumor cell escape from complement surveillance [80,81,82,83]. However, to date no studies on the pathobiological role of CD59 in EVs or its potential clinical utility in cancer detection has been reported. 

Tetraspanins are a vast superfamily of glycoproteins whose tetraspanin-enriched microdomains engage in a variety of specific molecular interactions. Among the tetraspanins previously identified on a variety of EVs are CD9, CD63, and CD81 [84]. We found 11 tetraspanins (TSPAN1, TSPAN5, TSPAN6, TSPAN8, TSPAN9, TSPAN14, TSPAN15, CD9, CD81, CD82, and CD151) in the EV proteome of at least three CRC cell lines and detected CD63 in the EV proteome of Colo205 and SW620 cell lines (Appendix A). TSPAN9, a 4-transmembrane protein involved in tumor growth and signal transduction, has been shown to be linked to tumor invasion, metastasis, and autophagy. The understudied gene encoding this tetraspanin was reported to be amplified in serous fallopian tube cancer [85]. Previous studies also revealed that high expression of TSPAN9 is an independent prognostic factor for poor overall survival of patients with gastric cancer [86] and showed that TSPAN9 is overexpressed in 5-fluorouracil (5-FU)-resistant gastric cancer cells, where it contributes to the chemoresistance to 5-FU by promoting autophagy [87]. However, there has been no research on the expression of TSPAN9 in human CRC tissue or tissue-derived EVs. In this context, we are also the first to explore the clinical significance of plasma EV levels of CD59 and TSPAN9 in cancer. In addition to identification of plasma EV CD59 and TSPAN9 as a two-marker panel for CRC detection, the pathobiological roles of EV CD59 and TSPAN9 in the tumor microenvironment and/or chemoresistance of CRC obviously remain as intriguing issues that warrant further investigation.

Through in-depth analysis of EV protein profiles from four CRC cell lines and bioinformatics annotation of the identified proteins, we have generated a focused EV membrane protein dataset containing 518 ‘core’ proteins released in common from at least three CRC cell lines (Appendix A). We found that 16.4% (754/4586, HT29), 20.5% (666/3243, SW480), 22.9% (503/2196, SW620) and 15.8% (780/4924, Colo205) of the identified EV proteins were classified as membrane proteins (Figure 3). The average across cell lines was 18.9%, which is close to the estimated value obtained from an analysis of proteomic data collected from the Vesiclepedia database in which 627 of 3027 evaluable EV proteins (21%) were classified as plasma membrane proteins [88]. By combing the EV proteome MS dataset with topological information on EV membrane proteins obtained by analysis using Protter software, we were able to map the location(s) of MS-detected peptide(s) in the amino acid sequences of three types of membrane proteins: single-transmembrane domain-containing proteins, multiple-transmembrane domain-containing proteins, and anchored proteins (Figure 3D and Appendix A). This information is very useful for designing workable targeted MS assays for detection and quantification of these EV membrane proteins in different biological or body fluid samples from distinct diseases or under different experimental conditions. 

Our study used differential ultracentrifugation to isolate EVs from cell conditioned medium and plasma, which has its own lacunae. Although ultracentrifugation can yield significant quantities of highly pure exosomes, it is unsuitable for clinical diagnosis owing to its low throughput, which only allows for the identification of six samples at a time, and poor reproducibility, making study-to-study comparability suspect. Ultracentrifugation paired with density gradient centrifugation (DGC) can improve exosome purity; however, DGC is time-consuming (centrifugation for 16–90 h), limiting its usefulness in clinical settings. A recent development of a magnetic bead-based EV enrichment approach for automated and high-throughput processing of body fluid samples in 96-well plates may represent an alternative purification strategy to achieve high throughput [89]. Another aspect is to increase the sample size to obtain more reliable and accurate mean values, uncover outliers that could bias data in a smaller sample, and achieve a lower margin of error. Furthermore, the inclusion of a maximum number of candidates in analysis pipelines can increase the likelihood of discovering therapeutically important indicators. In future studies, we would seek to increase case study numbers and include more targets for verification from our 2D-LC MS/MS and iTRAQ findings and possibly use a more sensitive and higher throughput MS assay, such as multiple-reaction monitoring (MRM) or parallel reaction monitoring (PRM) MS.

## 5. Conclusions

The present study explored the EV membrane protein profile in common among CRC cell lines and assessed alterations in the plasma EV proteome of CRC patients compared with healthy subjects. It further identified the plasma EV membrane proteins, CD59 and TSPAN9, as a novel biomarker panel that effectively detected CRC through targeted MS-based quantification of roughly a dozen selected candidate membrane proteins in plasma EV samples from 73 CRC patients and 80 age/sex-matched HCs. Because the samples used in this study were from a single hospital and the sample size was not large, the validity of this plasma EV biomarker panel should be validated using larger numbers of samples collected prospectively from multiple hospitals.

## Figures and Tables

**Figure 1 cancers-15-00177-f001:**
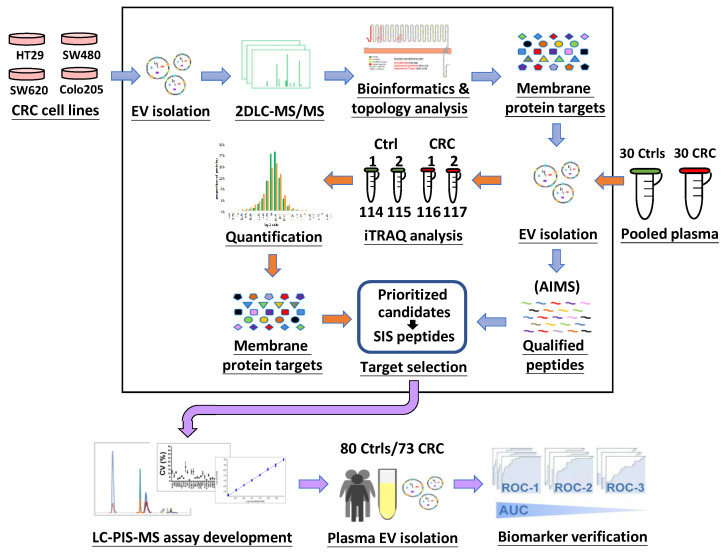
General design of the study. See text for details.

**Figure 2 cancers-15-00177-f002:**
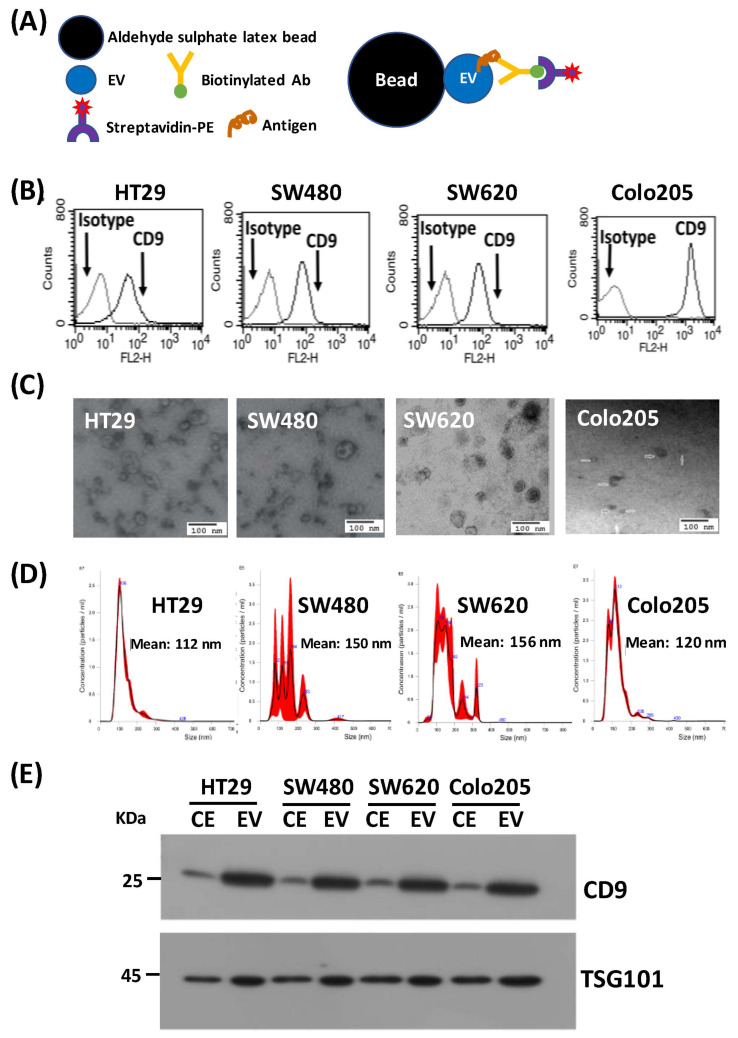
Characterization of EVs. (**A**) Principle of flow cytometry (FACS) analysis of EVs. (**B**) FACS analysis of CD9-positive EVs (30 μg EVs + 10 μL beads) released from HT29, SW480, SW620 and Colo205 cell lines. Mouse IgG1K was used as an isotype control. (**C**) Transmission electron microscopy (TEM) analysis of EVs isolated from conditioned media from the four CRC cell lines. (**D**) Determination of the size distribution of EVs isolated from conditioned media of the four CRC cell lines by nanoparticle tracking analysis (NTA). (**E**) Western blot analysis of the EV markers, CD9 and TSG101, in total cell extracts (CE) and EV fractions from the four CRC cell lines.

**Figure 3 cancers-15-00177-f003:**
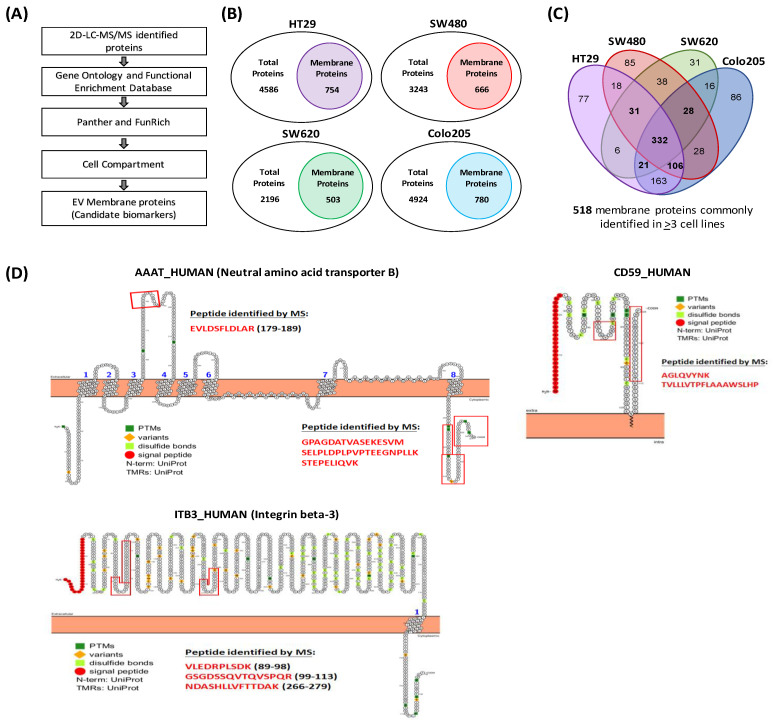
Identification of EV membrane proteins. (**A**) Workflow for the identification and selection of EV membrane proteins. The identified proteins from LC-MS/MS were sorted with Panther (Gene Ontology) and Funrich using Exocarta and Vesiclepedia databases for cell compartmentalization. (**B**) Total MS-identified proteins and membrane proteins associated with each of the four CRC cell lines, as predicted by GO databases. (**C**) Venn diagram showing the overlap of EV membrane proteins found in the four CRC cell lines. (**D**) Selected examples for topological analysis of target membrane proteins. MS-detected peptides for each identified protein are denoted by red boxes.

**Figure 4 cancers-15-00177-f004:**
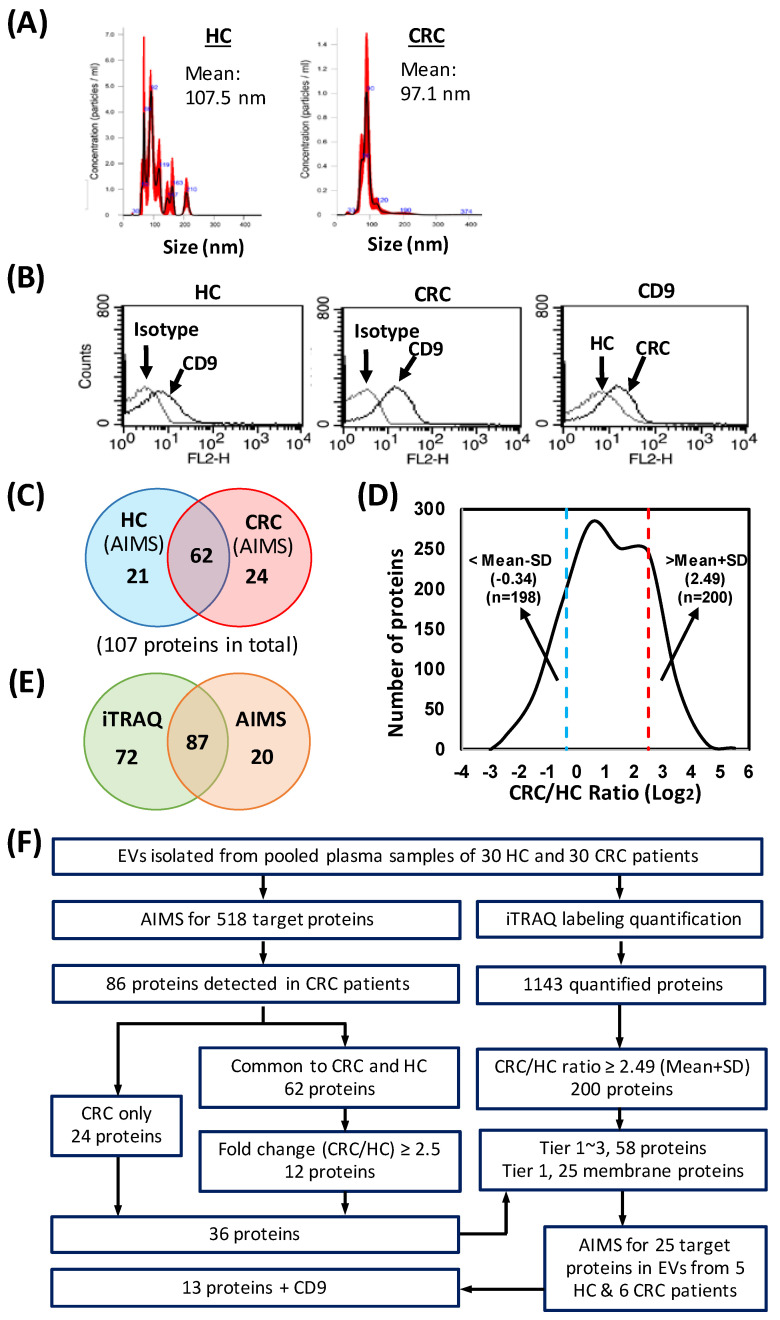
AIMS- and iTRAQ-based proteomic analysis of EVs prepared from plasma samples pooled respectively from HCs and CRC cases (n = 30/group). (**A**,**B**) Nanoparticle tracking analysis (**A**) and FACS analysis (**B**) of EVs isolated from pooled HC and CRC cases. Mouse IgG1K was used as an isotype control in FACS analysis. (**C**) Venn diagram showing proteins detected by AIMS in EVs isolated from pooled HC and CRC cases. (**D**) Log2 ratio distribution of the 1143 plasma EV proteins differentially expressed between CRC and HC groups, quantified by iTRAQ-based proteomic analysis. Dashed lines indicate boundaries at means ± SD. (**E**) Venn diagram showing the number of iTRAQ-identified proteins and the number of proteins common to AIMS and iTRAQ identification from the list of target proteins selected for AIMS. (**F**) Flow chart of the selection and categorization of prioritized candidates for further verification by targeted MS.

**Figure 5 cancers-15-00177-f005:**
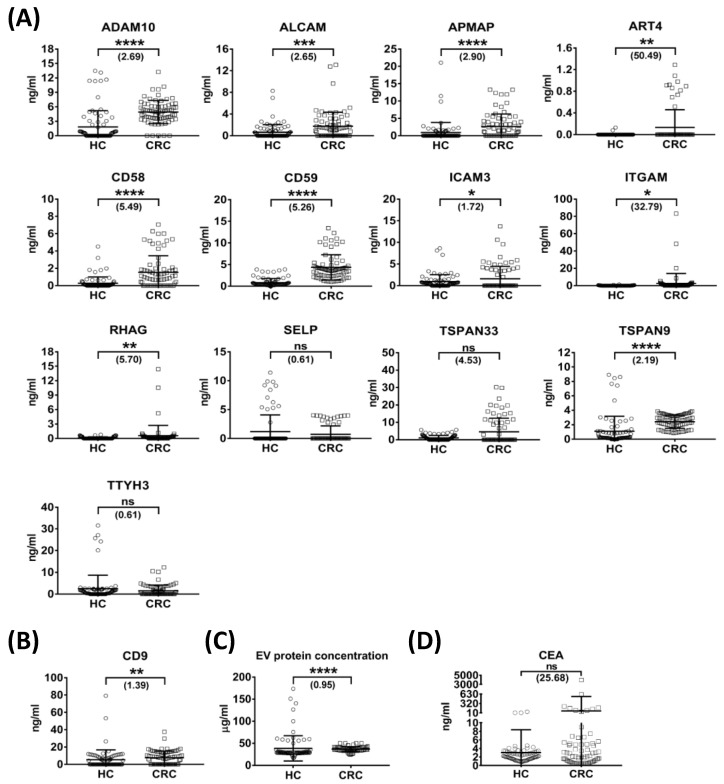
Measurement of protein levels of 13 targets and CD9, and EV total protein concentration, in plasma EVs from enrolled subjects. (**A**–**C**) Levels of 13 target proteins (**A**), CD9 (**B**), and EV total protein concentration (**C**) in plasma EV fractions from HCs and CRC patients. (**D**) Levels of CEA measured in plasma specimens from HC and CRC patients. Horizontal lines indicate means ± S.D. (* *p* ≤ 0.05; ** *p* ≤ 0.01; *** *p* ≤ 0.001; **** *p* ≤ 0.0001; ns, not significant).

**Figure 6 cancers-15-00177-f006:**
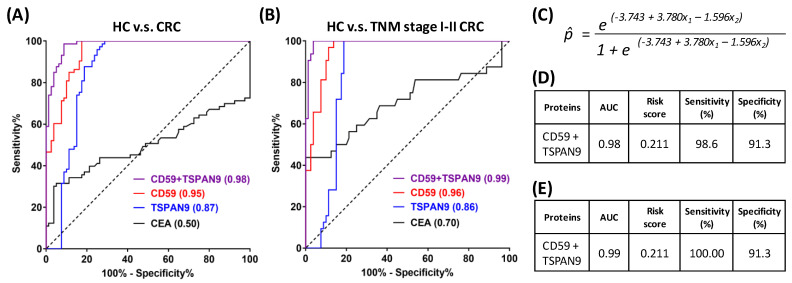
ROC curve analysis of CD59 and TSPAN9 in discriminating CRC patients from HC subjects. (**A**) The discriminatory power of plasma EV CD59, TSPAN9, and CD59 combined with TSPAN9 was evaluated in all HC subjects and CRC patients using ROC curve analysis. (**B**) The discriminatory power of these targets was further evaluated in all HC subjects and in TNM stage I and II CRC patients. (**C**) The equation for probability in risk evaluation was generated by logistic regression. (**D**,**E**) AUC values, risk score, sensitivity and specificity of CD59 and TSPAN9, used as a biomarker panel to distinguish CRC (**D**) or TNM stage I/II CRC (**E**) patients from HC subjects.

**Table 1 cancers-15-00177-t001:** List of candidate proteins selected for a verification study in individual samples by targeted MS assay.

Gene Name	Protein Name	Surrogate Peptide Sequence	Precursor mz	
			Heavy	Light
ADAM10	Disintegrin and metalloproteinase domain-containing protein 10	AIDTIYQTTDFSGIR	855.932	850.928
ALCAM	CD166 antigen	VLHPLEGAVVIIFK	514.988	512.316
APMAP	Adipocyte plasma membrane-associated protein	GLFEVNPWK	549.297	545.29
ART4	Ecto-ADP-ribosyltransferase 4 (CD297)	FGQFLSTSLLK	624.857	620.850
CD58	Lymphocyte function-associated antigen 3	VAELENSEFR	602.297	597.293
CD59	CD59 glycoprotein	AGLQVYNK	450.755	446.748
CD9	CD9 antigen	EVQEFYK	475.739	471.732
ICAM3	Intercellular adhesion molecule 3	IALETSLSK	485.289	481.281
ITGAM	Integrin alpha-M	LFTALFPFEK	610.843	606.836
RHAG	Ammonium transporter Rh type A	FLTPLFTTK	538.317	534.310
SELP	P-selectin	NEIDYLNK	508.760	504.753
TSPAN9	Tetraspanin-9	EGLLLYHTENNVGLK	854.461	850.454
TSPAN33	Tetraspanin-33	DDLDLQNLIDFGQK	821.414	817.407
TTYH3	Protein tweety homolog 3	VLHPLEGAVVIIFK	768.327	763.323

**Table 2 cancers-15-00177-t002:** Concentrations of 14 target proteins in plasma EV samples, as well as CEA in plasma samples, from HCs and CRC patients.

Protein	HC (n = 80)		CRC (n = 73)		CRC vs. HC				
	ng/mL *^a^*	Detectable *^b^*	ng/mL *^a^*	Detectable *^b^*	Fold Change *^c^*	*p*-Value *^d^*	AUC	Sensitivity (%)	Specificity (%)
ADAM10	1.83 ± 3.37	40/80	4.92 ± 2.41	68/73	2.69	<0.0001	0.83	93.15	77.50
ALCAM	0.66 ± 1.38	31/80	1.76 ± 2.58	43/73	2.65	0.0005	0.65	56.16	76.25
APMAP	0.91 ± 2.91	21/80	2.64 ± 3.62	40/73	2.90	<0.0001	0.67	39.73	92.50
ART4	0.00 ± 0.02	2/80	0.13 ± 0.33	11/73	50.49	0.0027	0.56	15.07	100.00
CD58	0.28 ± 0.72	31/80	1.54 ± 1.92	48/73	5.49	<0.0001	0.72	60.27	90.00
CD59	0.83 ± 0.94	77/80	4.35 ± 2.93	73/73	5.26	<0.0001	0.95	100.00	82.50
CD9	5.50 ± 11.29	36/80	7.66 ± 7.65	46/73	1.39	0.0035	0.63	53.42	73.75
ICAM3	0.93 ± 1.61	52/80	1.59 ± 2.86	22/73	1.72	0.0424	0.59	69.86	65.00
ITGAM	0.08 ± 0.22	16/80	2.72 ± 11.37	23/73	32.79	0.0142	0.59	31.51	98.75
RHAG	0.10 ± 0.21	18/80	0.58 ± 2.14	33/73	5.70	0.0059	0.61	43.84	82.50
SELP	1.19 ± 2.89	13/80	0.72 ± 1.45	15/73	0.61	0.8257	0.51	20.55	83.75
TSPAN33	0.99 ± 1.45	37/80	4.50 ± 7.81	23/73	4.53	0.9226	0.50	28.77	100.00
TSPAN9	1.11 ± 2.06	69/80	2.42 ± 0.90	73/73	2.19	<0.0001	0.87	100.00	71.25
TTYH3	2.52 ± 6.19	54/80	1.54 ± 2.60	31/73	0.61	0.1323	0.57	57.53	67.50
CEA	3.03 ± 5.35	80/80	77.75 ± 469.49	73/73	25.68	0.9906	0.50	31.51	95

***^a^*** Mean ± SD; ***^b^*** Detectable (concentration > 0) case number/total case number; ***^c^*** Fold change of protein levels in CRC group over healthy control (HC) group; ***^d^*** Mann–Whitney test and statistically significant at *p*-value < 0.05.

**Table 3 cancers-15-00177-t003:** Correlations of plasma EV ADAM10, CD59 and TSPAN9 levels with clinicopathological characteristics of CRC patients.

Characteristics	Case No.	ADAM10 (ng/mL)	*p*-Value	CD59 (ng/mL)	*p*-Value	TSPAN9 (ng/mL)	*p*-Value	CEA (ng/mL)	*p*-Value
Gender ***^a^***	-	-	-	-	-	-	-	-	-
Female	37	4.82 ± 2.62	0.9781	4.38 ± 3.11	0.9257	2.50 ± 0.91	0.3766	126.51 ± 657.7	0.9715
Male	36	5.02 ± 2.20	-	4.32 ± 2.77	-	2.33 ± 0.90	-	27.63 ± 58.38	-
Age (years) ***^a^***	-	-	-	-	-	-	-	-	-
<58 ***^c^***	34	4.69 ± 1.81	0.5629	4.29 ± 2.75	0.9494	2.38 ± 0.88	0.8798	10.64 ± 31.55	0.2391
≥58	39	5.12 ± 2.84	-	4.41 ± 3.1	-	2.45 ± 0.93	-	136.24 ± 639.71	-
Tumor stage ***^b^***	-	-	-	-	-	-	-	-	-
T1	14	4.39 ± 1.36	0.4949	3.39 ± 0.79	0.4428	2.07 ± 0.65	0.2655	0.88 ± 0.53	0.0035 ***^d^***
T2	9	5.14 ± 3.63	-	3.63 ± 2.72	-	2.25 ± 0.84	-	23.06 ± 60.24	-
T3	42	5.15 ± 2.29	-	4.81 ± 3.28	-	2.62 ± 0.90	-	112.46 ± 612.04	-
T4	8	4.40 ± 3.04	-	4.45 ± 3.42	-	2.15 ± 1.21	-	91.53 ± 214.20	-
Lymph node metastasis ***^a^***	-	-	-	-	-	-	-	-	-
N0	37	5.02 ± 2.22	0.0789	3.96 ± 1.83	0.1842	2.24 ± 0.71	0.0011 ***^d^***	6.16 ± 26.38	0.0003 ***^d^***
N1	36	4.82 ± 2.62	-	4.76 ± 3.72	-	2.60 ± 1.05	-	151.32 ± 664.63	-
Distant metastasis ***^a^***	-	-	-	-	-	-	-	-	-
M0	57	4.84 ± 2.51	0.2103	4.13 ± 2.84	0.0475 ***^d^***	2.30 ± 0.86	0.0104 ***^d^***	6.97 ± 29.67	0.0252 ***^d^***
M1	16	5.18 ± 2.05	-	5.15 ± 3.19	-	2.85 ± 0.94	-	329.87 ± 983.81	
TNM stage ***^b^***	-	-	-	-	-	-	-	-	-
Stage I	18	4.87 ± 2.42	0.3214	3.50 ± 0.73	0.2273	2.15 ± 0.64	0.0065 ***^d^***	1.40 ± 1.67	0.0010 ***^d^***
Stage II	14	4.85 ± 2.20	-	3.37 ± 0.86	-	2.13 ± 0.63	-	2.33 ± 1.91	-
Stage III	25	4.83 ± 2.82	-	5.00 ± 4.07	-	2.50 ± 1.07	-	13.59 ± 44.37	-
Stage IV	16	5.18 ± 2.05	-	5.15 ± 3.19	-	2.85 ± 0.94	-	329.87 ± 983.81	-

***^a,^******^b^****p*-values were generated using ***^a^*** Mann–Whitney test (mean ± SD) or ***^b^*** Kruskal–Wallis test (mean ± SD). ***^c^*** The threshold of age was determined by the median of all patients’ age. ***^d^*** Statistically significant, *p*-value ≤ 0.05.

## Data Availability

The mass spectrometry proteomics data have been deposited to the ProteomeXchange Consortium via the PRIDE [90] partner repository with the dataset identifier PXD038871.

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
