# Peer review of "Extracellular Vesicle Membrane Protein Profiling and Targeted Mass Spectrometry Unveil CD59 and Tetraspanin 9 as Novel Plasma Biomarkers for Detection of Colorectal Cancer"

_cancers, 2022, doi:10.3390/cancers15010177_

Round 1

Reviewer 1 Report

The manuscript entitled ” Extracellular vesicle membrane protein profiling and targeted mass spectrometry unveil CD59 and tetraspanin 9 as novel plasma biomarkers for detection of colorectal cancer ” has been carefully refereed. The authors have explored the EV membrane protein profile in common among colorectal cancer (CRC) cell lines and assessed alterations in the plasma EV proteome of CRC patients compared with healthy subjects. This is also the first report for identifying CD59 and TSPAN9 as novel biomarkers for CRC. The basic idea and concept of the paper are clearly described. This is a well-written paper. This paper is OK and publishable.

Reviewer 2 Report

In this work, authors performed 2D-LC-MS to explore the EV membrane protein profile in CRC cell lines, and validated these protein profiles in pooled plasma-generated EVs from CRC patients compared with healthy controls. Through these works, authors suggested a novel biomarker panel consisting of EV membrane proteins CD59 and TSPAN9 for detecting early-stage CRC. This work is potentially interesting, providing valuable clinical information in the field. There are several issues needed to be addressed before the publication.

1.    At line 515, authors suggest the EVs from CRC cells have similar molecular and physicochemical properties to EVs from other cells types. Authors need to be cautious or careful claim these, unless they have EVs from other cell types included in those analyses or at least have cited references about it. Otherwise, author might need to remove it.

2.    In Figure 2E, multiple bands are observed in CD63, which is very confusing. CD63 band is not completed for CE control in HT29 cell. For publication, authors needs to provide higher quality of WB images, and they could try different primary antibodies for CD63 as well, for example : https://www.abcam.com/cd63-antibody-late-endosome-marker-ab216130.html

3.    In the last paragraph of the discussion section, It is good that author recognized the limitation using ultracentrifugation for EVs purification with low throughput. However, this is not important to the whole story, especially author mentioned the six samples at a time which is more directly related to the UC equipment. Instead, author could discuss other potential alternative purification strategies to achieve high throughput.

Reviewer 3 Report

Re: Dash et al. Extracellular vesicle membrane protein profiling and targeted mass spectrometry unveil CD59 and tetraspanin 9 as novel plasma biomarkers for detection of colorectal cancer

The authors isolated Extracellular Vesicles (EV) from four colorectal cancer cell lines by ultracentrifugation and analyzed the proteome by mass spectrometry. Validation was performed on clinical material and targeted mass spectrometry. Results were associated with clinical data. 

The study is well-written and presents clinical valuable results. However, the experimental setup and the presentation of the data show only some major/minor shortcomings:

Major/minor considerations

1.     Did the authors use a loading control for the Western Blotting results in Figure 2? Which antibody was used for TSG101? 

2.     Please update Figures 3 & 4 as some text passages are not readable. 

3.     Did the author use a correction for multiple testing to detect differentially expressed proteins? Please add a clustering (PCA, tSNE, …) approach to visualize the reproducibility of the mass spectrometric analysis

4.     Did the authors exclude patients with chemotherapy, 30-days lethality, and R1/R2 resected patients? 

5.     How did the authors define healthy controls? 

6.     Why did the authors make their protein data not publicly available?

7.     The authors should carefully revise their manuscript regarding typos and grammar.

Round 2

Reviewer 3 Report

While the manuscript has been improved in many points, minor considerations still need to be addressed: 

- Please add lot and/or order information of the used antibodies in the main manuscript. 

- Please add exclusion criteria and the therapy data of stage 4 colorectal cancer patients also in the manuscript. 

- Please add the definition of healthy controls in the manuscript. 
